# Status and Numbers of the Brown Bear (*Ursus arctos* L.) in Bulgaria

**DOI:** 10.3390/ani13081412

**Published:** 2023-04-20

**Authors:** Ruslan Serbezov, Nikolai Spassov

**Affiliations:** 1National Museum of Natural History, Bulgarian Academy of Sciences, 1000 Sofia, Bulgaria; rserbezov@abv.bg; 2Institute of Information and Communication Technologies, Bulgarian Academy of Sciences, 1113 Sofia, Bulgaria

**Keywords:** *Ursus arctos* L., Brown Bear numbers, Bulgaria, Balkans

## Abstract

**Simple Summary:**

The article focuses on the conservation status and trends in the numbers of the bear in Bulgaria. Genetic research shows that the Bulgarian population, together with other Balkan bears and the bears from the Apennines, have a common and unique gene pool. This and their limited numbers make them particularly important from the point of view of conservation of the biodiversity of the bear in Europe. Our analysis shows that from the end of the 1980s, when the Bulgarian population was the most numerous (around and even over 700 bears), a trend of decreasing numbers began, which is due to human impact. The present study is based on the National Field-monitoring of the species from recent years, analyzed by using statistical methodology. According to our analysis, at present the population numbers about 500 individuals.

**Abstract:**

Until recently, the Bulgarian bear population (*Ursus arctos* L.) was considered one of the significant ones in Europe and one of the few with more than 500 bears. While the numbers of some neighboring populations may be increasing, the Bulgarian population has been on a downward trend since the early 1990s. The probable numbers of the species at the end of the 1980s was about 700–750 individuals. Calculations based on field data from national monitoring and statistical analysis show probable numbers in Bulgaria in 2020 of about 500 individuals (data for the autumn state). This decline is mostly related to poaching due to weaker control activity, the reduction of forest areas and habitat fragmentation. The preservation of the Bulgarian population, which, together with the other Balkan populations and the Apennine bear, has a unique gene pool, is particularly important from the point of view of preserving the biodiversity of the species in Europe.

## 1. Introduction. Distribution, Numbers and Characteristics of the Balkan and Bulgarian Population of the Brown Bear

According to IUCN Red List data (see: Brown Bear (*Ursus arctos*)-IUCN Red List https://www.iucnredlist.org) (accessed on 20 December 2022), the brown bear is found in 22 European countries. The following populations can be distinguished (according to the IUCN Red List, with some of our interpretations): Russian-Finnish, Swedish-Norwegian, Carpathian, Dinaric-Pindus, Eastern Balkan (mainly in Bulgaria), Alpine, Abruzzo, Cantabrian, and Pyrenean. The largest is the Russian population, and, in the EU, the Carpathian population.

Two main subpopulations can be distinguished within the Balkan meta-population [1,2,3]. Dinaro-Pindus population: This population, according to the IUCN Red list, is of about 3940 bears. They are spread over nine countries, covering an area of 115,300 sq. km. The numbers are as follows: Slovenia (564), Croatia (937), Bosnia and Herzegovina (1000), Montenegro (378), North Macedonia (375), Albania (190), Serbia (120), Kosovo (unknown), and Greece (450–500). We consider the quoted data in some of the countries to be overestimated when the inhabited area [1] is taken into account.

The Eastern Balkan population: mostly in Bulgaria, in three segments with around 421 bears, but also in Greece and Serbia with around 42 (25–98) and 3–5 bears, respectively, this population inhabits about 39,000 sq. km. The population is considered stable with 613 (468–665) bears, although estimates fluctuate according to the evaluation methods. The Greek part of the Rila–Rhodopes segment is near the Dinaro–Pindus population and recently, a possible connection between those two populations has been noted [4]. On the northeast of the Balkan Mountain segment, there is a potential, unproven relation with the Carpathian population.

After the 1930s in Bulgaria, the bear inhabited only the mountains, mostly the Middle Balkan mountain Range and the Rila–Rhodope massif. In the former location, the population extends over more than 120 km., stretching along the main ridge from the Zlatishko–Tetevenski region to the Elensko–Tvardishki one. Individual bears reach Mount Etropolska Baba in the western part of the ridge. In its eastern part, there are sporadic observations in the area of Varbishki Pass. In the Rila–Rhodope massif, the bear lives in Rila, Pirin, Slavyanka, and in the Western Rhodopes. A small part of the latter population extends to Greece, inhabiting the southern slopes of the Western Rhodopes and the Slavyanka mountain. Small populations exist in Vitosha and the surrounding small mountains, as well as in Sredna Gora, Kotlenska mountain, and the Western Balkan Range (Western Stara Planina). They are characterized by low density, unstable structure, and sporadic reproduction, compared to the populations in Rila-Rhodope massif and in the Middle Balkan Range. Individual animals (including females with cubs) have been observed in the mountains along the western border of Bulgaria (Osogovo, Karvav Kamak, Maleshevska mountain regions), and in the Eastern Rhodopes. Their presence is related to wandering individuals, feeding migrations, and temporary habitats [2,5,6,7,8] (Figure 1).

Until recently, the Bulgarian population has been considered as one of the most significant in Europe and one of the few with more than 500 bears. By the national law on Biodiversity, the bear is a protected species. It is also categorized as a threatened species in the Red Data Book of Bulgaria: VU. criteria: D1 [6]. As of 2008, a Bulgarian bear management plan is functioning, and is being regularly updated.

Genetic studies show that two genetically distinct clades (mitochondrial haplogroups) of the brown bear occur in Europe: one in southern Europe (clade 1), and the other in the entire north-eastern part of the continent (clade 3) [9]. Clade 1 is the only lineage in the European Mediterranean since the beginning of the Holocene [10], and the only one in Bulgaria since the end of the Pleistocene [11]. The Balkan-Apennine population is unique in its gene pool (clade 1b) [11]. Until recently, this uniqueness was not taken into account when assessing the conservation significance of bear populations in Europe. The uniqueness of the gene pool increases the conservation significance of the Balkan-Apennine population. This concerns the Balkan metapopulation and its fragments, to which the Bulgarian population belongs [12]. The numerical size of the population is a main indicator of its status.

The assessment of the recent Bulgarian bear population is rather controversial (Table 1). For this reason, the aim of the present research is to provide a sufficiently reliable estimate of the numbers of bears in the country, based on a statistical analysis of the field data from the national monitoring of the species.

## 2. Materials and Methods. Methodology for Assessment of the Bear Population

### 2.1. Filed Data from the Annual National Field Monitoring of the Brown Bear

The numbers of the brown bear population in Bulgaria is based on field data, statistically presented as samples (see Chapter 2.2). They were collected via the route method. The bear signs recorded included primarily footprints (width of fore paw footprint and length of hind paw footprint (Table 2), but also excrements, and tree-marks (see [24,25]). The field data were gathered annually as part of monitoring, organized by the Executive Environment Agency (EEA) at the Ministry of the Environment and Water (MOEW) of Bulgaria. The route method is described in the ‘Methodology for monitoring of the brown bear’ as part of The National Biodiversity Monitoring System of the EEA (https://eea.government.bg/bg/bio/nsmbr/praktichesko-rakovodstvo-metodiki-za-monitoring-i-otsenka/UrsusArctos_MetodikaMonitoring.pdf) (accessed on 17 March 2023). The methodology for differentiating of the footprints by sex and age is described in [24]. The monitored territories covered the habitats of the bear in the country, distributed in trial areas (squares—10 × 10 km). In them, suitable routes have been chosen for finding traces of the species’ vital activity (Figure 2A–D).

### 2.2. Statistical Assessment of the Numbers of the Bear Population in Bulgaria: Methodology

The statistical analysis of the brown bear population numbers allowed us to evaluate the numbers according to the types of habitats (see below); to evaluate the numbers of each geographically separate territory (mountain region); to evaluate the numbers at national level (the overall area of distribution of the species in the country); to evaluate the age structure; and to evaluate the average density of the population in the studied trial areas (the density was indicated for the purposes of also assessing the status of the species in the Alpian and Continental biogeographic regions).

The habitats of the brown bear can be classified into five types, using the classification as defined in CORINE land cover-2018 (https://www.eea.europa.eu/publications/COR0-landcover) (accessed on 17 March 2023). In this way, the following types of forests can be distinguished as parts of the trial areas (squares of 10 × 10 km):Deciduous forests (311) (the number is after the CORINE classification)Coniferous forests (312)Mixed forests (313)Vegetation communities of shrubs and forests (322) and transitional woody-shrub (324)Others—the sum of the area of all other types of forest from the habitat of the brown bear as per CORINE land, covering the square.

As noted, the field data presented as samples, include primarily footprints (Table 2), but also other signs (see Chapter 2.1). Identical or similar footprints, found in close distance to one another, could be selected using a software application. The software application ‘Assessment of the status of the brown bear population in Bulgaria based on mathematical, statistical, and biological analysis from the monitoring’, in accordance with Contract No. 3682/13.12.2018, financed by the Enterprise for management of environmental protection activities (EMEPA) at MOEW, allowed calculation of the numbers of the brown bear population, using the Bootstrap methodology [26]. This method allowed assessment of the population on the basis of data (primarily unique signs) from defined territory and on the basis of statistical calculation by using replicated data from new unique signs, provided by subsequent monitoring in the same area. Identical or similar traces were eliminated.

The first step in the assessment process was the identification of the unique footprints based on field data collected during the national monitoring. The numbers of unique footprints detected was determined by experts using the developed software product. After the determination of the unique footprints, the program automatically distributed them in the corresponding 5 types of forests, as well as in the residual area. At this stage, the program made a statistical assessment of the numbers of the brown bears in the area of distribution of the bear in Bulgaria. When not all of the planned 240 routes (included in the 142 grids covering the bears‘ habitats) were traversed during fieldwork, the program automatically divided the number of grids into two groups. The first group included those networks that contained transects (grids) visited during the national monitoring. Brown bear numbers were statistically estimated using the maximum likelihood method. The second group included those networks that contained transects (grids) not visited during the national monitoring. The population numbers in the second set were obtained by extrapolation.

When calculating the population of the brown bear, maximum plausibility was used. The coefficients *λ*_1_, …, *λ*_5_ were introduced to assess the probability of finding a bear in a certain area (type of forest), representing parameters of the Poason distribution. These parameters defined the numbers of bears per 1 sq.km in the relevant type of forest. Additionally, the coefficients *ψ*_1_, …, *ψ*_5_ were introduced to calculate the probability of find a bear in case it inhabits the respective area (square) and type of forest.
(1)Pi,j,d=e−λj∑k=d∞kdψid(1−ψj)k−dψjd(1−ψj)k−dλjkk!

Formula (1) represents the function of maximum plausibility. When $\psi_{j}=1$ (i.e., in the cases that we have feeds for wild animals in each square), a simpler formula is achieved: $e^{-\lambda_{j}}\frac{{\lambda_{j}}^{d}}{d!}$, where *d* is the number of the unique routes. By using the method of maximum plausibility, the value of the coefficients $\lambda_{1}$, …, $\lambda_{5}$ and $\psi_{1}$, …, $\psi_{5}$ could be assessed.

## 3. Results

### 3.1. Number of Bears in Bulgaria. Review and Analysis of Data since the First Half of 20th Century (Table 1, Figure 3)

Based on the first official research for the species (a survey from 1934), the number of bears amounted to 300–366, with 32 bears killed per year [13]. Around the middle of the last century, when the distribution was similar to the current one, the poll data revealed about 450 bears in 1959 [14,27]. Even if we consider that the estimated data based on conducted surveys were not very precise, they provide decent estimations for the numbers of the population numbers in the first half of 20th century. They also show a trend of increasing numbers due to measures taken to protect the species. Based on the assessment of the population from the late 1970s, the maximum number at that time was around 600 bears [5,15]. According to a survey from the mid-1980s [19], the number amounted to 700–750 bears, and then about 750 at the end of the 1980s [20]. Based on the analysis from the late 1990s, there were probably under 700 bears inhabiting Bulgaria at that time [2]. It should be noted that the data from the official taxation data from the 1980s [16,17], and those from the beginning of the 21st century, [21] given by the Ministry of Forestry and the Ministry of Agriculture, were not based on scientific methods, and we consider them to be overestimated. In any case, despite the probable approximation of the data, we cannot doubt the visible upward trend of the bear population over time from 1930s to 1980s. This trend was due to the measures taken for the protection of the species at the beginning of this period, as well as to the fact that bears were treated as hunted species in the 1970s and 1980s. Consequently, a number of measures have been taken to increase the population: artificial feeding, artificial breeding, and the release of young bears into nature (mainly from the Carpathian population), as well as the strong restriction on poaching. All these measures have led to reaching the maximum of population numbers for the last 100 years. At the same time, this has also led to some negatives for the population effects (overpopulation in several regions and the genetic contamination of the unique European biodiversity of the local population (see below)).

**Figure 3 animals-13-01412-f003:**
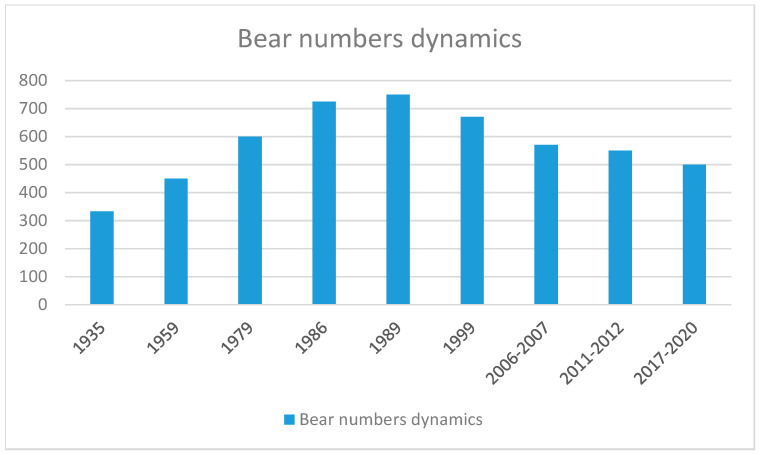
Dynamics of the brown bear population numbers in Bulgaria from the 1930s to 2020. (Data from the sources mentioned in Table 1. The official data from the Ministry of the Forestry and the Ministry of Agriculture were not used: see the text).

The data for the end of 20th century and the beginning of 21th century are rather controversial. Due to increased poaching related to weaker control as a result of social changes, the population decreased in the 1990s [2,28]. According to official taxation data from the Ministries of Agriculture and Forest, the population numbers for the 2001–2004 period were between 800 and 900 bears with a clear trend for permanent increase [21]. These data seem rather overestimated (see above). The analysis of population numbers performed during the preparation of the Red Book of the Republic of Bulgaria (2015; electronic edition 2011), based on the existing data for the density of the species in different biotopes and on the data for the home range size and for the entire area of distribution, shows numbers for the 2006–2007 period of about 550 bears, of which 150–190 were in the Middle Balkan range, 300–350 in the Rila–Rhodope massif and 25–30 bears in the other habitats of the country [6,7]. The follow-up of the analyses and of the data for the population numbers showed that weakening of control lead to a decrease in the numbers of the species in the country from the 1990s onwards. After a clear decrease at the end of 20th, and the beginning of 21st century, in the 2006–2007 period and in 2012 the numbers of the population seemed to stay at a constant level. This is evident from the data of the EEA at the MOEW (2022) [22], received from the national monitoring. In a report on the brown bear under the Nature 2000 project, the population numbers are estimated for 2012 at 470–540 bears [7]. According to us, in that analysis the data in the Rhodope habitat are underestimated (part of the habitats in the border areas are not included in the analysis), so the approximate expert assessment (500–600 bears), mentioned in the same source, seems more realistic.

Another assessment from the same period (2012) gave similar results: a population of 530–590 bears (as part of the east-Balkan population) within a total area of 18,900 sq. km (permanent habitat) and 2100 sq. km (occasional habitat), with the population categorized as stable [3]. Against the background of these data, the number of 600–800 bears for the same period (2012), given by another source [29], seems rather exaggerated. In relation to a genetic analysis of the population for the period 2008–2012, about 368 samples of brown bear were gathered and analyzed, mainly from the region of the Rhodopes and to a lesser extent from the Balkan Mountain. Based on 13 microsatellites, in total 136 unique individuals were identified. The sex was defined based on the mitochondrial DNA and as a result, two genetic lines were proven. Based on these data, the population of bears in the Rhodopes was assessed at 206–334 individuals [23]. The statistical error in this case was substantial due to the low numbers of the analyzed sample. The lower limit of the estimation seems more realistic, considering the previous expert assessments [2,6].

According to the data from the National Biodiversity Monitoring System of EEA (2022) [22], obtained during the field monitoring by applying a national methodology (see Chapter 2.1) over a 10-year period, the numbers of the bear population in the period 2011–2014 varied between 518 and 523 individuals. From 2015 (439 individuals) onwards the number constantly decreased year by year, reaching 321 bears by 2020. The national monitoring did not cover all bear habitats and the collection and analysis of the footprints did not manage to report all traces from the monitoring period. Analysis of the EEA according to the national monitoring appeared to underestimate the numbers of the bear population, which was likely due to the reasons mentioned above. Despite the issues with the accuracy of the stated bear population numbers, the data for the 10-year period reveals a clear trend of a decrease in the numbers of bears in the recent years.

### 3.2. Statistical Assessment of the Current Numbers of the Bear Population in Bulgaria

By applying the mentioned in chapter 2 algorithm for calculating the population in different habitats, integrated in the software application and based on activity signs recorded during the performed monitoring program (2017–2020), the following results for bear numbers in the country are shown Table 3):

The number of years during which the indicated data were collected is too small to be able to trace any trend. It should be taken into account that not all footprints were identified during the field monitoring. At the same time, within the monitoring territory, covering 127 trials, each year there are several unvisited trials (different according to the years) where small numbers of uncounted bears remain. Apart from that, there are territories which are inhabited by several bears which are not included in the monitoring route. Such uncounted bears inhabit the regions of the Western Balkan mountain range, Osogovo, the mountainous area around Trun, the area to the west of Tryavna mountain, and the Eastern Rhodopes. There are also several wandering individuals. All mentioned-above cases were not included in the field monitoring. As a result, the actual numbers of bears in the country is higher. In our expert opinion, the non-counted bears in the monitoring assessment are almost 100 per year, so the numbers of the bear population in the last years is around 500 per year. These bears inhabit territory of 12,826.185 sq. km. [7], of which 3199.93 sq. km. (approximately 1/3) are the territories of national and nature parks.

## 4. Discussion. Status of the Population and Its Conservation Significance

In summary, we can state again that despite some uncertainty in the published data, a visible increase in the numbers of the species was observed between the thirties and the end of the 1980s (see Table 1). Indeed, the practice from the 1970s and 1980s of breeding and releasing young bears, mainly Carpathian, into the wild (in the Rhodopes and in Central Stara Planina), the different gene pool, had a negative effect on the purity of the autochthonous population. Fortunately, that seems to only have slightly affected [30] the genetic purity of the population. From the beginning of 1990s until 2005-2007 population numbers decreased noticeably. As it seems, this negative trend remains until now (Table 1). Our claim is that current population numbers are close to 500 individuals (see Section 3.2). This is a higher number than the one quoted by EEA—320–350 individuals. Even in case the number we quote is not quite precise, it speaks about a population decrease by more than 30% for the last 35 years, whereas the decrease in the last 5–6 years could be higher. The bear population has thus reached that of the 1950s–1960s, and this gives cause for concern about the current state of the species in Bulgaria.

The decreasing population numbers is related primarily to poaching, due to less active controlling, to the decrease in forest territories, and to the fragmentation of the habitats [6,8]. According to some opinions, the frequent cases of bears entering the mountain villages indicate an increase in the number of certain local bear populations. We cannot agree with such a statement. The mentioned phenomenon has completely different reasons, mostly related to the depopulation of the villages, leading to a lack of fear of humans and their activities; to the feeding of game, provided by local hunters near the mountain villages; and the indiscriminate dumping of food waste in the areas of the species’ habitats.

As we have already noted, new research shows that the Balkan-Apennine population is unique in its gene pool [11]. This specific genome apparently accounts for the phenotypic features of this population. It is distinguished by its coloration, in which light-brown (‘golden’) individuals are strongly represented (a general characteristic of the Mediterranean bear), and by the presence of significant sexual dimorphism in coloration (light-brown (‘golden’) coloration being characteristic of female individuals) [1,25]. The indicated differences between the Mediterranean bear and the bears from the rest of Europe suggest a need for revising the taxonomic status of this population, and certainly give grounds for raising the conservation status of the southern (Mediterranean) European population [1,12].

## 5. Conclusions

The Bulgarian bear population represents the main part of the Eastern Balkan population [3]. The analysis showed, despite the possible inaccuracy of data, a clear trend of increasing numbers of the bear population from the 1930s to the end of the 1980s, due to the efforts made for conservation (see above). This analysis also showed that reduced control due to socio-economic changes since the early 1990s has led to a reverse trend after the end of the 1980s. This decline was mostly related to an increased in poaching due to weaker control, to reduction of the suitable forest territories in the mountains, and to habitat fragmentation. After a peak of the population in the late 1980s of probably up to 750 bears (according to the more reliable data available), this number began to decline. According to current research, up to 500 bears inhabit the country (2020, data for the autumn state). This finding is a cause for concern.

Above, we have drawn attention to the uniqueness of the gene pool and the specific morphological characteristics of the Balkan-Apennine bear, part of which is the Bulgarian population. Until now, these features have not been taken into account when assessing the conservation significance of regional bear populations, but there is no doubt that they reinforce the need to protect these populations. From this point of view, maintaining an optimal population in Bulgaria (where 1/3 of the population lives in optimal conditions and protected natural areas) is of particular importance for the conservation of the biodiversity of the species in Europe. The current updating of a new 10-year. action plan for the species focuses specifically on measures to develop and maintain the bear population in Bulgaria.

## Figures and Tables

**Figure 1 animals-13-01412-f001:**
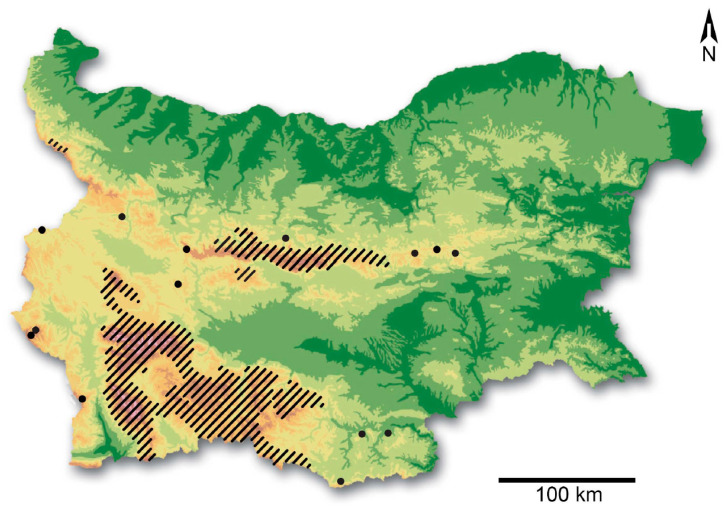
Recent distribution of the brown bear in Bulgaria (from sources: [2,6,7], with additions). Hachured areas: presence of local subpopulations, dots: single specimens/wandering individuals/temporary (periodic) appearances/seasonal sporadic appearance. The northern hatchured area represents the distribution of the bear in the Balkan mountain range, and the southern in the Rila-Rhodopean mountain massif. The brown colors correspond to the mountainous regions.

**Figure 2 animals-13-01412-f002:**
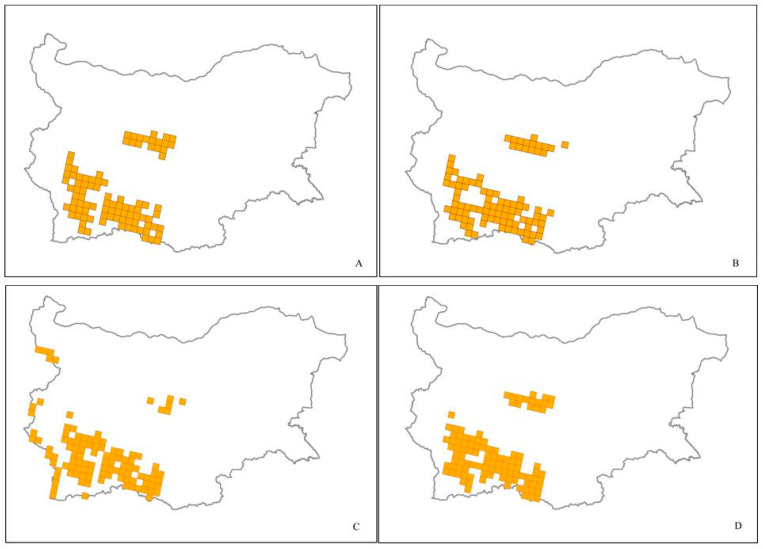
Territory from the national field monitoring of the bear in the main habitats of the species in Bulgaria, distributed in trial areas with a surface of 10 × 10 km. ((**A**–**D**) show the monitoring areas for 2017–2020, respectively).

**Table 1 animals-13-01412-t001:** Numbers of the Bulgarian bear population (for close to 100 years after different sources).

Numbers of the Main Local Populations			Total Numbers in Bulgaria	References
Rilo–Rhodope Massif	Rhodope Mountains	Middle Balkan Range		
			**300–366** (for 1935)	[13]
			**450** (for 1959)	[14]
		**100** (adults)	Up to **600** (for 1979)	[5]
			**600** (for 1979)	[15]
			About **850** (for 1982–1983 after Ministry of forestry)	[16]
			**847** (for 1983 after official data of the Ministry of Forestry)	[17]
		**86** (for the southern slopes; 1987)		[18]
			**700–750** (for 1986)	[19]
**500–520**		**200–210**	**750** (end of 1980s)	[20]
			**Less than 700** (end of 1990s)	[2]
			**800–900** (for 2001–2004, after official data of the Ministry of Agriculture)	[21]
	Less than **200**	Less than **200**	**550** (for 2006)	[2]
			**500–700** (for 2007)	[8]
300–360		**150–190**	**ca. 530**	[6]
			**600–800** (for ~2012)	[22]
			**500–600** (for 2011–2012)	[7]
	**206–334** (for 2009–2012, but see the text)			[23]
**255**	**186**	**65**	**332** on average per year (for 2017–2020)	[22]
upper limit **375**	upper limit **237**	upper limit **79**	Up to/around **500** on an average per year (for the 2017–2020 period)	this study

**Table 2 animals-13-01412-t002:** Size of the footprints of the brown bear from Bulgaria. Congruence between the length of the footprint of the hind paw, the width of the fore paw, and the size, sex, and age of the bear (from source [25]).

Feature Category Bear	Width of the Fore Paw Footprint	Width of the Hind Paw Footprint	Length of the Hind Paw Footprint
**1**. A bear cub—1st year	5–7 cm	-	6–11 cm
**2**. A bear cub—2nd year, up to ~50 kg.	~8–9\10	0–0.5 cm narrower than the anterior one	12–15
**3**. Young females (3 and 4 year) and young males ~three-years old (small bear: ~50–100 kg)	10/11–12 cm. (Most frequent in the field). Young individuals of 12 cm are probably young males as 12 cm is a normal size for a mature female)	0–0.5 cm narrower than the anterior	16–19/20
**4**. Adult females and subadult (four- or five-year old) males (average-sized bear—100~200 kg.)	11–12/13–13.5/14; (the second digits are mostly for males; bears of 13 cm are rarely a female but most frequently a young male)	~0.5–1 cm narrower than the anterior	19/20–23/24 cm; (23/24—only male individuals)
**5**. Mature males more than 5 years old (large bear ~200–250 kg)	14/14.5–17	It could be up to 1–1.5 cm narrower	24–26/27 cm
**6**. Very big, old males, usually more than 10 years old and more than 250 kg (records—above 350 kg)	17 and more	Up to 1–2 cm narrower	27–30 (31?) cm

**Table 3 animals-13-01412-t003:** Estimated total population numbers of brown bears in Bulgaria for the period 2017–2020.

Year	Average	Lower Limit	Upper Limit
2017	401	338.24	459.83
2018	374	320.39	416.69
2019	397	344.71	445.83
2020	457	360.64	551.69
2017–2020	407	340.995	468.51

## Data Availability

Part of data reported in this study are contained within the article. The raw data used in the study are from the database of the EEA at the MOEW of Bulgaria and are publicly available upon request.

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
