# Peer review of "Status and Numbers of the Brown Bear (Ursus arctos L.) in Bulgaria"

_animals, 2023, doi:10.3390/ani13081412_

Round 1

Reviewer 1 Report

Dear Authors

The manuscript submitted for review is interesting, but needs improvement. I have a few comments. Add brown bear in the title to clarify what species of bear the authors are referring to. The Discussion section should be rewritten, there is actually repeated information from the Simple Summary or Abstract. This is by far the weakest part of the paper, perhaps because the authors were overly modest in their use of data from the cited literature. The References section lists only 23 literature references, 12 of which have one of the authors of this manuscript as first author or co-author. The Conclusions section should be rewritten.

Regards

Reviewer 2 Report

Studies of this type that monitor poulations of threatened species over time are a critical part of informed conservation policy development and practice.  The work is thus not highly novel but is an important contribution.

  1. the Research addresses the status of a population of a threatened species 
  2. The work is highly relevant to species conservation but as it forms part of decadal monitoring it is not and should not be novel in the sense of new discovery.
  3. It adds to knowledge by continuing the assessment of population numbers which appear to be declining.
  4. Methodology is appropriate
  5. Conclusions are consistent with the data
  6. References appropriate
  7. Figures are appropriate.

Author Response

Please see the attacment

Reviewer 3 Report

Authors

Your study includes important results regarding the current status (2017-2020) of the black bear populations in Bulgaria. These results, when compared with reports of the past century, show an alarming decrease of the population size. Given the low total numbers, a scientific based report is essential to incorporate this pertinent information properly into the conservation management plans currently being developed. However, the way in which this paper is now written does not hold up to scientific standards. The methods appear sound, but results aren't presented in a clear fashion. I would recommend a table showing the estimated population sizes in the past century for the different regions. This, for instance, would make for much easier reading. Right now, particularly because of the poor English, I am easily lost as I read the results. Moreover I would recommend a figure, which would demonstrate the change in population size over the past century. Sorry to say but it is obvious that your English is clearly that of non-native speakers. Thus, it is essential, given that the majority of sentences in this manuscript (MS) are very awkward, that you have the entire manuscript cleaned up by a professional English writer, and, particularly, someone with experience in scientific publications. I have included some suggestions, but can not do the entire MS for you. Sorry. Please go through the attached pdf, in which I have made numerous comments to help you improve the manuscript before your resubmission to this or another journal. Good luck with your outstanding work on the brown bear.

General

In results section, you mention several times that past population numbers are “overstated”. I assume you mean “overestimated”.

You often make comments that are subjective and thus you need to explain clearly why you feel these past reports are overestimating the populations.

An example is found in the discussion section:

Today the numbers of the population (in our opinion it is higher than the one quoted by EEA – 320-350 individuals) is towards/around 500 220 individuals.

Introduction

You have not included what your objectives are for this research study. This usually is explained in the last paragraph of the introduction.

Methods

You mentions the monitoring method to be in some government manual but no link or citation, plus probably not too accessible so you need to go into describing this method in this paper.

Round 2

Reviewer 1 Report

The authors have incorporated all the suggested corrections. Thank you.

Regards

Author Response

Dear Reviewer,

We would like to thank you for accepted revisions of the second version of the manuscript. According to your suggestion to correct the English language, we used the help of person with fluent English to prepare the current (second) revision of the manuscript.

Sincerely,

Nikolai Spassov,

corresponding author